# SKELETON-TO-IMAGE ENCODING: ENABLING SKELETON REPRESENTATION LEARNING VIA VISION-PRETRAINED MODELS

## ABSTRACT

Recent advances in large-scale pretrained vision models have demonstrated impressive capabilities across a wide range of downstream tasks, including cross-modal and multi-modal scenarios. However, their direct application to 3D human skeleton data remains challenging due to fundamental differences in data format. Moreover, the scarcity of large-scale skeleton datasets and the need to incorporate skeleton data into multi-modal action recognition without introducing additional model branches present significant research opportunities. To address these challenges, we introduce Skeleton-to-Image Encoding (S2I), a novel representation that transforms skeleton sequences into image-like data by partitioning and arranging joints based on body-part semantics and resizing to standardized image dimensions. This encoding enables, for the first time, the use of powerful vision-pretrained models for self-supervised skeleton representation learning, effectively transferring rich visual-domain knowledge to skeleton analysis. While existing skeleton methods often design models tailored to specific, homogeneous skeleton formats, they overlook the structural heterogeneity that naturally arises from diverse data sources. In contrast, our S2I representation offers a unified image-like format that naturally accommodates heterogeneous skeleton data. Extensive experiments on NTU-60, NTU-120, and PKU-MMD demonstrate the effectiveness and generalizability of our method for self-supervised skeleton representation learning, including under challenging cross-format evaluation settings.

## 1 INTRODUCTION

Recent years have witnessed the remarkable success of large-scale vision-pretrained models, such as Vision Transformers (ViTs) (Dosovitskiy et al., 2021), Masked Autoencoders (MAE) (He et al., 2022), and Vision-Language Models (VLMs) (Jia et al., 2021; Singh et al., 2022), across diverse visual recognition and understanding tasks. These models leverage abundant image data to learn transferable representations and are increasingly adapted to other modalities, including depth maps (Xia & Wu, 2024; Yang et al., 2024b), IR images (Zhang et al., 2023; Paranjape et al., 2025), video sequences (Tong et al., 2022; Wang et al., 2023; 2022a), and even point clouds (Wang et al., 2022b; Zhang et al., 2022b). A common strategy is to project non-image modalities into 2D formats, enabling direct use of image-pretrained models. While this works for dense 3D data like point clouds, applying similar projections to 3D skeleton data poses unique challenges. Skeletons are inherently sparse (only 15-30 joints per frame) and exhibit articulated structures with strong semantic relationships—unlike unstructured point clouds. Moreover, skeleton sequences span temporal dimensions critical for motion understanding, making naive 2D projections inadequate.

While direct application of vision models to skeleton data is non-trivial, skeleton-based representation learning remains fundamental for understanding human motion. Skeleton data provides a compact, appearance-invariant, and high-level abstraction of human activities, making it particularly valuable for tasks such as

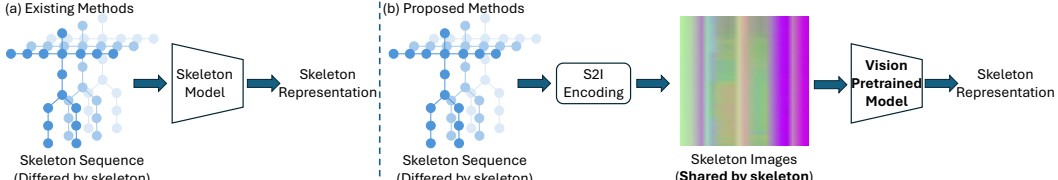

Figure 1: The existing methods train the skeleton models directly, while the proposed method converts skeleton data into image-like data and then train with the pre-trained vision models.

action recognition, gait analysis, and human-computer interaction. Furthermore, as multi-modal action recognition gains traction, skeleton data serves as a complementary modality that can enhance robustness and interpretability when effectively integrated with dense visual inputs like RGB images and depth maps. However, the scarcity of large-scale annotated skeleton datasets and the incompatibility of skeleton structures with vision model architectures limit current methods' generalization ability across diverse tasks and scenarios.

To overcome these challenges, we introduce an approach that leverages pretrained Vision Models—such as MAE (He et al., 2022) and DiffMAE (Wei et al., 2023)— for skeleton representation learning. This extends the use of vision models beyond 2D images to the 3D skeleton domain by transferring rich knowledge from large-scale image pretraining. At the core of this approach is our proposed **Skeleton-to-Image Encoding (S2I)**, a novel representation method that reformats skeleton sequences into image-like representation compatible with vision models. Specifically, the 3D joint coordinates $(x, y, z)$ are directly mapped to RGB channels, converting motion patterns into pseudo-images. To ensure semantic consistency, we first partition skeleton joints into five body parts: torso, left arm, right arm, left leg, and right leg. These are then reordered by following the body part sequence, and within each part, joints are further sorted in a top-down manner based on their physical positions. We then stack these reordered joints across the temporal dimension, producing a spatial-temporal image-like representation of the entire skeleton sequence. Finally, the generated representation is resized to the standard image input size ($224 \times 224$) for compatibility with vision models. As a result, our method enables, for the first time, the direct application of powerful pretrained vision models for self-supervised skeleton representation learning, effectively transferring rich visual domain knowledge to the skeleton domain without requiring task-specific architectural modifications. A visual comparison between existing skeleton pipelines and our approach is shown in Figure 1.

Current skeleton-based methods are typically designed for homogeneous skeleton formats, relying on fixed joint definitions and dataset-specific architectures. Such designs limit their scalability and hamper adaptation to skeleton data with varying joint configurations, coordinate systems, or capture devices. As a result, these methods struggle in cross-format scenarios, where skeleton layouts differ significantly across datasets. In contrast, our proposed S2I provides a unified and format-agnostic representation. By abstracting skeleton data into a consistent image-like structure, our method naturally supports joint training across multiple heterogeneous skeleton datasets, facilitating universal skeleton representation learning. Using diverse skeleton formats collectively, our approach enhances model generalization and captures richer motion dynamics.

Through extensive experiments on benchmark datasets, including NTU-60, NTU-120, and PKU-MMD, we demonstrate that our method achieves competitive performance in self-supervised skeleton representation learning. Notably, our approach shows exceptional effectiveness in challenging cross-format evaluation, exhibiting strong generalizability across heterogeneous skeleton formats.

Our contributions can be summarized as follows: (1) We propose a novel pipeline that leverages vision-pretrained models and their weights for skeleton-based representation learning, bridging the modality gap between images and skeleton sequences. (2) We introduce Skeleton-to-Image Encoding, a unified representation method that reformats sparse 3D skeleton data into image-like inputs, compatible with vision models and resilient to skeleton format variations. (3) We explore our method's effectiveness for heterogeneous skeleton representation learning and propose a universal skeleton pretraining strategy, enabling cross-format learning across multiple skeleton datasets.

## 2 RELATED WORKS

**Skeleton Action Recognition.** In recent years, deep learning has been widely used in skeleton action recognition, owing to its strong capability in feature extraction and representation learning. Existing methods typically adopt Recurrent Neural Networks (RNNs)(Du et al., 2015b; Liu et al., 2017; Zhang et al., 2017), Convolutional Neural Networks (CNNs)(Du et al., 2015a; Ke et al., 2017; Li et al., 2018), Graph Convolutional Networks (GCNs)(Yan et al., 2018; Shi et al., 2019; Chen et al., 2021), and Transformers-based models (Zhang et al., 2021b; Zhou et al., 2022) to directly learn skeleton representations. However, most existing methods are specifically designed for homogeneous skeleton formats, assuming fixed joint numbers. Such designs inherently limit their scalability and generalization to datasets with diverse skeleton structures. In contrast, our proposed S2I representation offers a format-agnostic solution that naturally handles heterogeneous skeleton layouts, enabling more flexible and robust skeleton representation learning.

**Self-Supervised Skeleton Representation Learning.** Self-supervised skeleton representation learning has become a promising direction to reduce reliance on costly manual annotations. Recent methods mainly adopt contrastive learning (Li et al., 2021; Guo et al., 2022; Zhang et al., 2022a) and masked modeling (Mao et al., 2023; Wu et al., 2024; 2023) to learn discriminative features from unlabeled skeleton data. In contrast, large-scale pretrained vision models have shown remarkable transferability across diverse downstream tasks and even different modalities. However, leveraging such pretrained models for skeleton data remains largely unexplored due to the inherent gap in data formats. In this work, we propose S2I, which reformats skeleton sequences into image-like inputs, enabling the direct use of vision-pretrained models for skeleton tasks.

**Self-Supervised Representation Learning on Images.** Self-supervised learning has gained significant traction in computer vision for its ability to learn effective representations without manual annotations. Contrastive learning methods exploit augmentation invariance to learn instance-discriminative features from images and videos (He et al., 2020; Chen et al., 2020; Qian et al., 2021). More recently, masked modeling has emerged as a powerful alternative to contrastive methods. MAE (He et al., 2022) reconstructs masked pixel values using an asymmetric encoder-decoder architecture, providing a simple yet effective framework for visual representation learning. BEiT (Bao et al., 2022) follows a mask-then-predict strategy, using visual tokens generated by a pre-trained tokenizer as prediction targets. DiffMAE (Wei et al., 2023) further enhances MAE by introducing a denoising diffusion process to iteratively reconstruct masked regions. In this work, we evaluate our proposed S2I representation using both MAE and DiffMAE pretrained models.

## 3 METHOD

### 3.1 SKELETON-TO-IMAGE ENCODING

As discussed in Section 1, our objective is to leverage vision-pretrained models for skeleton representation learning. This enables the effective utilization of large-scale vision models and their pretrained weights for skeleton-based tasks. Furthermore, adopting a unified vision model backbone facilitates multi-modal action recognition, where diverse data modalities can be seamlessly integrated.

To achieve this, skeleton data must be reformatted into a representation compatible with image-based models. Specifically, it is essential to encode spatial-temporal information from skeleton sequences in a form analogous to image data, thereby enabling knowledge transfer from pretrained vision models. However, a fundamental challenge arises from the inherent differences in data structures. While image data is typically represented as tensors of size $3 \times 224 \times 224$ for vision models, skeleton sequences are structured as $T \times J \times 3$, where $T$ denotes the temporal length of the sequence, and $J \times 3$ represents the 3D coordinates $(x, y, z)$ of skeleton joints. To bridge this discrepancy, we propose a straightforward yet effective mapping strategy. The 3D joint coordinates $(x, y, z)$ are directly assigned to the RGB channels of an image, with each spatial axis corresponding to one color channel. The remaining challenge is to project the $T \times J$ spatial-temporal data into the $224 \times 224$ image plane, ensuring compatibility with standard vision model inputs.

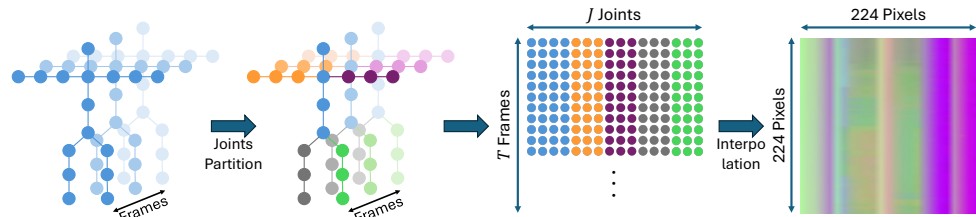

Figure 2: Illustration of the Skeleton-to-Image Encoding (S2I) process, which transforms skeleton sequences into image-like representations via joint partitioning, temporal stacking, and interpolation.

To bridge the gap between skeleton sequences and image-based model inputs, we propose **Skeleton-to-Image Encoding (S2I)**, a novel representation that transforms skeleton sequences into dense, image-like data compatible with vision models (e.g., MAE, DiffMAE), as illustrated in Figure 2. Specifically, we partition the human skeleton into five semantic body parts: spine, left arm, right arm, left leg, and right leg. This strategy, widely adopted in prior works (Li et al., 2020; Cai et al., 2019; Zhang et al., 2021a), follows the kinematic structure of the human body and can be generalized across skeleton formats.

Within each part, joints are arranged along their kinematic chain, ordered by distance from the torso. For example, joints in the left leg are sequenced as: left hip → left knee → left ankle → left foot. This ordering preserves the spatial relationships inherent in the skeleton structure. For temporal modeling, the 3D positions of each joint across $T$ frames are stacked to form a spatio-temporal feature map of size $T \times J$, where $(x, y, z)$ coordinates are assigned to the RGB channels of the image. Finally, to match the input requirements of vision models (e.g., $224 \times 224$), we apply linear interpolation along both temporal and joint dimensions, resizing them independently to a fixed resolution of $224 \times 224$. This yields an image-like representation of the skeleton sequence while preserving essential spatial-temporal patterns.

**Application & Advantage.** Our S2I reformats skeleton sequences into image-like representations that are inherently robust to variations in skeleton structures. By abstracting skeleton data into a unified format, it enables seamless representation of diverse skeleton layouts without relying on dataset-specific joint definitions. In contrast, conventional skeleton-based methods are tightly coupled to homogeneous skeleton formats, assuming fixed joint numbers, which limits their scalability across datasets with differing skeleton structures. Even recent works (Yang et al., 2021b; Liu & Wang, 2022) that attempt cross-format evaluations still rely on shared joint subsets, fundamentally adhering to homogeneous representations.

Our approach differs by introducing a format-agnostic representation paradigm. This design decouples skeleton representations from dataset-specific joint configurations, enabling direct integration of skeleton data from multiple heterogeneous sources. As a result, S2I naturally supports **cross-format representation learning**, where models trained on one skeleton format can generalize to others. Furthermore, it facilitates **universal representation pretraining** by jointly leveraging diverse skeleton datasets, similar to practices in large-scale image pretraining, without requiring task-specific architectural modifications. A visual comparison between conventional skeleton pipelines and the two proposed settings is provided in Appendix A.4.

### 3.2 VISION REPRESENTATION MODELS

Our proposed S2I reformats skeleton data into image-like data, enabling seamless application of vision-pretrained models for skeleton representation learning. Unlike skeleton networks, our approach leverages the general-purpose architecture and pretrained weights of powerful vision models.

In this work, we evaluate two representative vision models to demonstrate the effectiveness of our representation: MAE (He et al., 2022) and DiffMAE (Wei et al., 2023). Both models are originally designed for image representation learning and pretrained on large-scale ImageNet datasets, providing rich visual priors that can be effectively transferred to the skeleton domain through our unified representation. **MAE** learns

visual representations by reconstructing masked image patches from the visible context. Leveraging our S2I encoding to convert skeleton sequences into image-like data, we reuse ImageNet-pretrained MAE weights and perform skeleton pretraining via masked reconstruction. **DiffMAE** further enhances this framework by incorporating iterative denoising processes inspired by diffusion models. Following the same strategy, we utilize ImageNet-pretrained DiffMAE weights and conduct skeleton pretraining using our S2I representation.

The skeleton-pretrained models are evaluated on downstream skeleton tasks to validate the effectiveness of our proposed S2I representation. While we focus on MAE and DiffMAE in this work, our S2I representation is compatible with a broad range of vision models, including emerging generative and multimodal architectures.

## 3.3 MASK SAMPLING STRATEGY

The effectiveness of masked modeling largely depends on the masking strategy employed during pretraining. To optimize representation learning for skeleton data, we investigate several masking strategies applicable to both image-based and skeleton-specific contexts. **Random Masking** is the standard approach used in image-based masked modeling (He et al., 2022), where image patches are randomly masked without considering spatial relationships. Formally, given a mask ratio $r$, we randomly select $\lfloor r \times N \rfloor$ patches to mask from the total $N$ patches in the skeleton image representation. **Block Masking** increases pretraining task difficulty by masking contiguous regions of the input. Starting from randomly selected seed positions, blocks of adjacent patches are masked together, encouraging the model to learn stronger local structural relationships. Beyond these general strategies, we introduce two skeleton-specific masking schemes designed to better capture the unique spatial-temporal structure of human motion: **Joint Masking** focuses on the spatial domain by randomly masking joints across the skeleton body, challenging the model to infer missing joint positions based on articulated body structure. **Temporal Masking** targets the temporal dimension by masking entire frames or temporal slices, encouraging the model to capture dynamic motion patterns from partial sequences.

## 3.4 TRAINING OBJECTIVES

We adopt a two-stage training pipeline for skeleton action recognition, leveraging our Skeleton-to-Image (S2I) representation to adapt vision-pretrained models to the skeleton domain.

In the first stage, we perform *self-supervised skeleton representation learning* by applying masked modeling on our S2I representation, using the ImageNet-pretrained MAE (He et al., 2022) and DiffMAE (Wei et al., 2023) as backbones. Given a skeleton sequence $\mathbf{X} \in \mathbb{R}^{3 \times 224 \times 224}$ transformed via S2I, we apply masked modeling to learn skeleton representations: For MAE, we minimize the reconstruction loss between the original input and the reconstruction of the masked patches:

$$\mathcal{L}_{\text{MAE}} = \frac{1}{|\mathcal{M}|} \sum_{i \in \mathcal{M}} \left\| \hat{\mathbf{X}}_i - \mathbf{X}_i \right\|_2^2, \tag{1}$$

where $\mathcal{M}$ denotes the set of masked patches, $\hat{\mathbf{X}}_i$ is the reconstructed patch, and $\mathbf{X}_i$ is the ground truth. For DiffMAE, we follow (Wei et al., 2023) and reconstruct the masked regions through a denoising diffusion process, conditioned on the visible parts. The loss is defined as:

$$\mathcal{L}_{\text{DiffMAE}} = \mathbb{E}_{t,x_0,\epsilon} \left\| x_m^0 - D_\theta \left( x_m^t, t, E_\phi(x_v^0) \right) \right\|_2^2, \tag{2}$$

where $x_m^0$ is the original masked region, $x_m^t$ is the noised version at diffusion step $t$, $x_v^0$ is the visible region, $E_\phi$ denotes the encoder, and $D_\theta$ is the diffusion decoder.

In the second stage, we evaluate the skeleton-pretrained encoders on downstream *skeleton action recognition* tasks by attaching a classification head and optimizing with cross-entropy loss: $\mathcal{L}_{\text{CE}} = -\sum_{c=1}^{C} y_c \log \hat{y}_c$, where $C$ is the number of classes, $y_c$ is the ground-truth label, and $\hat{y}_c$ is the predicted probability. Depending on evaluation protocol, we either perform linear probing with a frozen encoder or fine-tune the entire model.

## 4 EXPERIMENTS

### 4.1 DATASETS

For evaluation, we conduct experiments on five datasets: NTU-60 (Shahroudy et al., 2016), NTU-120 (Liu et al., 2020), PKU-MMD (Liu et al., 2020), NW-UCLA (Wang et al., 2014), and Toyota (Das et al., 2019). The first three use 25-joint skeletons, while NW-UCLA and Toyota contain 20 and 13 joints, respectively. **NTU-60** is the most widely used benchmark for skeleton action recognition, comprising 56,880 skeleton sequences across 60 action classes performed by 40 subjects. In our experiments, we follow the standard cross-subject (C-sub) and cross-view (C-view) evaluation protocols. **NTU-120** is the largest skeleton-based action recognition dataset to date, containing 114,480 samples across 120 action classes, collected from 106 subjects across 32 setups with varying locations and backgrounds. We follow the official cross-subject (C-sub) and cross-setup (C-set) evaluation protocols for benchmarking. **PKU-MMD** is a large-scale benchmark for continuous multi-modality 3D human action understanding, featuring approximately 20,000 action instances in 51 categories. PKU-MMD consists of two subsets: Part I (easier, cleaner data) and Part II (challenging, noise data). We follow the cross-subject protocol for both subsets in our experiments. **NW-UCLA** contains 1,494 action samples from 10 classes, performed by 10 subjects and captured using three Kinect v1 cameras. Each skeleton consists of 20 joints. **Toyota** is a real-world dataset for daily living activity recognition, containing 16,115 videos across 31 classes. We follow the original evaluation protocols, conducting transfer learning experiments under both cross-subject (CS) and cross-view1 (CV1) settings.

### 4.2 EXPERIMENTAL SETUP

**Network Architecture.** For both MAE and DiffMAE, we adopt the ViT-B architecture as the encoder. The decoder consists of eight transformer blocks, each with a channel dimension of 512. In DiffMAE, the decoder follows a cross-self attention design inspired by the original Transformer encoder-decoder structure. Specifically, in each decoder block, the noised tokens first attend to the visible latent representations via a cross-attention layer, followed by self-attention among noise tokens to refine predictions. [1]

**Data Processing Details.** For NTU-60, NTU-120, PKU-MMD, and Toyota, we follow the data pre-processing in (Zhang et al., 2020), sequence level translation based on the first frame is performed to be invariant to the initial positions. If one frame contains two persons, it is split into two single-skeleton frames. For NW-UCLA, we adopt the data pre-processing in (Cheng et al., 2020).

**Implementation Details.** For the self-supervised skeleton representation learning stage, we initialize the models with ImageNet-pretrained MAE and DiffMAE weights, and adopt their default implementation configurations (He et al., 2022; Wei et al., 2023). For skeleton action recognition, we use SGD with Nesterov momentum (0.9) for linear probing, with an initial learning rate of 0.2 decayed via cosine annealing. For fine-tuning, we employ the AdamW optimizer with an initial learning rate of 0.001, decayed by cosine annealing. All skeleton action recognition experiments are trained for 100 epochs. The batch size is set to 128 for all datasets, except for NW-UCLA, where a smaller batch size of 32 is used. [1]

### 4.3 ABLATION STUDY

We conduct ablation studies on the **NTU-60 C-sub** to examine three key aspects of our method: (1) the effectiveness of Skeleton-to-Image representation, (2) the impact of image-pretrained weights, and (3) the choice of masking strategies and skeleton modalities.

**Effectiveness of Skeleton-to-Image Representation with Vision Models.** To evaluate the viability of using image-based models for skeleton representation learning, we evaluate MAE and DiffMAE architectures in combination with our proposed Skeleton-to-Image Encoding. As shown in Table 1, both models achieve strong performance after skeleton SSL pretraining, followed by linear probing or fine-tuning, demonstrating

---

[1]Detailed network configurations and more implementation details are provided in Appendix A.1, A.2, and A.3.

|  | Image Pretrain | Skeleton Pretrain | MAE | DiffMAE |
|---|---|---|---|---|
| Linear-Probe |  |  | 52.0 | 52.0 |
|  | ✓ |  | 72.2 | 71.3 |
|  |  | ✓ | 76.3 | 78.6 |
|  | ✓ | ✓ | 81.4 | **83.1** |
| Fine-tune |  |  | 82.8 | 82.8 |
|  | ✓ |  | 86.8 | 86.5 |
|  |  | ✓ | 88.1 | 88.6 |
|  | ✓ | ✓ | 90.5 | **91.0** |

Table 1: Ablation study of image pretrain, skeleton pretrain.

| Strategy | Ratio | Linear Probe | Fine-tune |
|---|---|---|---|
| Joint | 75 | 82.3 | 90.4 |
| Temporal | 75 | 82.9 | 90.7 |
| Random | 75 | **83.1** | **91.0** |
| Group | 75 | 81.3 | 90.3 |

Table 2: Comparison of masking strategies.

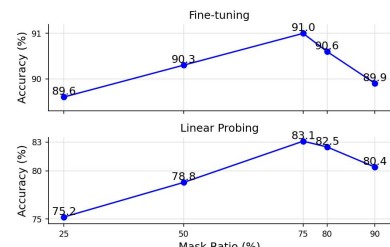

Figure 3: Effect of mask ratio

| Method | Linear Probe | Fine-tune |
|---|---|---|
| Joint | 83.1 | 91.0 |
| Motion | 70.5 | 89.7 |
| Bone | 81.0 | 90.8 |
| 3-stream | **85.8** | **93.1** |

Table 3: Comparison of skeleton modalities.

the viability of vision models for skeleton representation learning. These results demonstrate that, despite being designed for dense image data, vision models can effectively process structured skeleton sequences when equipped with appropriate representations. Skeleton-to-Image Encoding serves as an effective bridge, enabling the reuse of powerful vision models without task-specific architectural modifications.

**Impact of Image-Pretrained Weights.** We further examine the benefit of ImageNet-pretrained weights by comparing models trained from scratch and models initialized with pretrained weights. Table 1 shows that image-pretraining yields substantial gains in both linear probing and fine-tuning settings. In the linear probing scenario, pretrained MAE improves from 52.0% (scratch) to 72.2%, while pretrained DiffMAE improves from 52.0% to 71.3%. This substantial gap highlights the benefit of transferring generic visual representations to the skeleton domain. Furthermore, even after Skeleton Pretrain, pretraining with image data continues to provide notable gains, underscoring the importance of leveraging large-scale image-pretrained weights. These findings confirm that image-pretrained weights serve as a strong and transferable starting point for skeleton representation learning, significantly enhancing performance. Given DiffMAE's consistently superior performance, we adopt it as the **default backbone** for subsequent studies and comparisons.

**Effect of Masking Ratio.** We investigate the influence of different masking ratios using DiffMAE. As shown in Figure 3, increasing the masking ratio generally improves representation quality, with 75% yielding the best results in both linear probing and fine-tuning. Based on these findings, we adopt a **75%** masking ratio as the default setting for subsequent ablation studies and main comparisons.

**Masking Strategies.** We evaluate joint, temporal, random, and group masking strategies. Results in Table 2 indicate that random masking consistently outperforms others. While joint and temporal masking provide competitive results, group masking underperforms, suggesting that overly structured masking may limit the model's capacity. Consequently, we adopt **random masking with 75% ratio** as our default strategy.

**More Skeleton Modalities.** In skeleton-based recognition, joint data can be augmented by deriving motion and bone modalities. We evaluate these modalities individually and in combination using DiffMAE fine-tuning to assess their effectiveness within vision models. As shown in Table 3, joint features perform strongly, while motion and bone streams offer complementary cues. Fusing all three modalities yields substantial improvements, reaching 85.8% (linear probing) and 93.1% (fine-tuning). This confirms that multi-modality fusion remains effective even when skeleton data is reformatted as images and processed by vision models. In subsequent main results, we report the results of both the Joint stream (**S2I**) and the 3-stream fusion (**3s-S2I**).

## 4.4 COMPARISON WITH THE STATE-OF-THE-ART METHODS

**Linear Evaluation Results.** In linear evaluation, we freeze the pretrained backbone and train a supervised linear classifier on top. Table 4 shows results on NTU-60, NTU-120, and PKU-MMD. Despite using image-

| Method | NTU-60 | | NTU-120 | | PKU-MMD | |
|---|---|---|---|---|---|---|
| | C-sub | C-view | C-sub | C-set | Phase I | Phase II |
| LongT GAN (Zheng et al., 2018) | 39.1 | 48.1 | - | - | 67.7 | 26.0 |
| P&C (Su et al., 2020) | 50.7 | 76.3 | 42.7 | 41.7 | 59.9 | 25.5 |
| MS$^2$L (Lin et al., 2020) | 52.6 | - | - | - | 64.9 | 27.6 |
| SkeletonMAE (Wu et al., 2023) | 74.8 | 77.7 | 72.5 | 73.5 | 82.8 | 36.1 |
| 3s-SkeletonCLR (Li et al., 2021) | 75.0 | 79.8 | 60.7 | 62.6 | 85.3 | - |
| 3s-Colorization (Yang et al., 2021b) | 75.2 | 83.1 | - | - | - | - |
| ISC (Thoker et al., 2021) | 76.3 | 85.2 | 67.1 | 67.9 | 80.9 | 36.0 |
| GL-Transformer (Kim et al., 2022) | 76.3 | 83.8 | 66.0 | 68.7 | - | - |
| 3s-CrosSCLR (Li et al., 2021) | 77.8 | 83.4 | 67.9 | 66.7 | 84.9 | 21.2 |
| 3s-AimCLR (Guo et al., 2022) | 78.9 | 83.8 | 68.2 | 68.8 | 87.4 | 39.5 |
| CMD (Mao et al., 2022) | 79.4 | 86.9 | 70.3 | 71.5 | - | 43.0 |
| 3s-CPM (Zhang et al., 2022a) | 83.2 | 87.0 | 73.0 | 74.0 | 90.7 | 51.5 |
| 3s-ActCLR (Lin et al., 2023) | 84.3 | 88.8 | 74.3 | 75.7 | - | - |
| MAMP (Mao et al., 2023) | 84.9 | 89.1 | 78.6 | 79.1 | 92.2 | 53.8 |
| MacDiff (Wu et al., 2024) | **86.4** | **91.0** | **79.4** | 80.2 | **92.8** | - |
| S2I (Ours) | 83.1 | 88.0 | 75.0 | 75.5 | 88.0 | 57.4 |
| 3s-S2I (Ours) | 85.8 | 89.7 | 78.9 | **80.3** | 92.3 | **62.0** |

Table 4: Comparison of Linear Evaluation results on NTU 60, NTU and PKU datasets. 3s- represents the ensemble results of joint(J), bone(B) and motion(M) streams. **Bold** and underlined indicate the best and second best results, respectively. The same notation applies throughout.

| Method | NTU-60 | | NTU-120 | |
|---|---|---|---|---|
| | C-sub | C-view | C-sub | C-set |
| AimCLR (STTFormer) (Guo et al., 2022) | 83.9 | 90.4 | 74.6 | 77.2 |
| CrosSCLR (STTFormer) (Li et al., 2021) | 84.6 | 90.5 | 75.0 | 77.9 |
| CPM (Zhang et al., 2022a) | 84.8 | 91.1 | 78.4 | 78.9 |
| 3s-CrosSCLR (Li et al., 2021) | 86.2 | 92.5 | 80.5 | 80.4 |
| 3s-AimCLR (Guo et al., 2022) | 86.9 | 92.8 | 80.1 | 80.9 |
| SkeletonMAE (Wu et al., 2023) | 86.6 | 92.9 | 76.8 | 79.1 |
| 3s-Colorization (Yang et al., 2021b) | 88.0 | 94.9 | - | - |
| 3s-ActCLR (Lin et al., 2023) | 88.2 | 93.9 | 82.1 | 84.6 |
| MCC (Su et al., 2021) | 89.7 | 96.3 | 81.3 | 83.5 |
| ViA (Yang et al., 2024a) | 89.6 | 96.4 | 85.0 | 86.5 |
| 3s-Hi-TRS (Chen et al., 2022) | 90.0 | 95.7 | 85.3 | 87.4 |
| MAMP (Mao et al., 2023) | **93.1** | 97.5 | 90.0 | **91.3** |
| MacDiff (Wu et al., 2024) | 92.7 | 97.3 | - | - |
| S2I (Ours) | 91.0 | 96.7 | 86.6 | 87.9 |
| 3s-S2I (Ours) | **93.1** | **97.7** | **90.2** | 91.2 |

Table 5: Comparison of Fine-tuning results on the NTU-60 and NTU-120 datasets.

| Method | NTU-60 | | | |
|---|---|---|---|---|
| | C-sub | | C-view | |
| | (1%) | (10%) | (1%) | (10%) |
| LongT GAN (Zheng et al., 2018) | 35.2 | 62.0 | - | - |
| MS$^2$L (Lin et al., 2020) | 33.1 | 65.1 | - | - |
| ASSL (Si et al., 2020) | - | 64.3 | - | 69.8 |
| ISC (Thoker et al., 2021) | 35.7 | 65.9 | 38.1 | 72.5 |
| 3s-CrosSCLR (Li et al., 2021) | 51.1 | 74.4 | 50.0 | 77.8 |
| 3s-Colorization (Yang et al., 2021b) | 48.3 | 71.7 | 52.5 | 78.9 |
| CMD (Mao et al., 2022) | 50.6 | 75.4 | 53.0 | 80.2 |
| 3s-Hi-TRS (Chen et al., 2022) | 49.3 | 77.7 | 51.5 | 81.1 |
| 3s-AimCLR (Guo et al., 2022) | 54.8 | 78.2 | 54.3 | 81.6 |
| 3s-CMD (Mao et al., 2022) | 55.6 | 79.0 | 55.5 | 82.4 |
| CPM (Zhang et al., 2022a) | 56.7 | 73.0 | 57.5 | 77.1 |
| SkeletonMAE (Wu et al., 2023) | 54.4 | 80.6 | 54.6 | 83.5 |
| MAMP (Mao et al., 2023) | 66.0 | 88.0 | 68.7 | 91.5 |
| MacDiff (Wu et al., 2024) | 65.6 | 88.2 | 77.3 | **92.5** |
| S2I (Ours) | 71.4 | 84.8 | 73.3 | 87.8 |
| 3s-S2I (Ours) | **75.2** | **88.3** | **77.7** | 91.7 |

Table 6: Comparison of semi-supervised results on the NTU-60 dataset.

pretrained vision models without skeleton-specific architectures, our S2I achieves competitive performance compared to recent specialized methods. Specifically, S2I attains 83.1% and 88.0% on NTU-60 (C-sub and C-view), and 75.0% on NTU-120 (C-sub), demonstrating its effectiveness in bridging skeleton data with vision models. By integrating joint, motion, and bone streams, 3s-S2I further enhances representation quality, achieving state-of-the-art results on NTU-60 C-sub (85.8%), NTU-120 C-set (80.3%), and PKU II (62.0%).

**Fine-tuning Evaluation Results.** In Fine-tuning protocol, an MLP head is attached to the pre-trained backbone and the whole network is fully fine-tuned. As shown in Table 5, our S2I achieves competitive performance on NTU-60 and NTU-120 without skeleton-specific model designs. By bridging skeleton data with vision-pretrained models, S2I adapts effectively to skeleton tasks. With multi-stream fusion, 3s-S2I reaches state-of-the-art results, confirming the scalability of our approach.

**Semi-supervised Evaluation Results.** In semi-supervised protocol, both the classification head and the pretrained encoder are fine-tuned using only a small fraction of the training set. We evaluate on NTU-60 with 1% and 10% of the training set. As shown in Table 6, our S2I achieves strong performance with 71.4% (1%) and 88.4% (10%) under C-sub setting. With multi-stream fusion, 3s-S2I achieves state-of-the-art results, reaching 75.2% (1%) and 88.3% (10%), validating the effectiveness of S2I in low-label regimes.

| Method | To PKU-II | | |
|---|---|---|---|
| | NTU-60 | NTU-120 | PKU-I |
| LongT GAN (Zheng et al., 2018) | 44.8 | - | 43.6 |
| MS²L (Lin et al., 2020) | 45.8 | - | 44.1 |
| ISC (Thoker et al., 2021) | 51.1 | 52.3 | 45.1 |
| CMD (Mao et al., 2022) | 56.0 | 57.0 | - |
| SkeletonMAE (Wu et al., 2023) | 58.4 | 61.0 | 62.5 |
| MAMP (Mao et al., 2023) | 70.6 | 73.2 | 70.1 |
| MacDiff (Wu et al., 2024) | 72.2 | 73.4 | 71.4 |
| S2I (ours) | 70.2 | 71.6 | 68.1 |
| 3s-S2I (ours) | **72.6** | **73.9** | **72.9** |

Table 7: Comparison of transfer learning results on PKUMMD II dataset.

| Method | NTU-60 (25 joints) | | |
|---|---|---|---|
| | Toyota (13 joints) | | NW-UCLA |
| | CS | CV1 | (20 joints) |
| Colorization (Yang et al., 2021b) | - | - | 93.3 |
| UNIK (Yang et al., 2021a) | 63.1 | 22.9 | - |
| ViA (Yang et al., 2024a) | 64.5 | 36.1 | - |
| S2I (Ours) | 65.4 | 43.1 | 93.2 |
| 3s-S2I (Ours) | **70.1** | **53.8** | **94.2** |

Table 8: Cross-format transfer learning results from NTU-60 to Toyota and NW-UCLA datasets.

| Method | NTU120 C-sub | NTU60 C-sub | PKU-I | PKU-II | Toyota-CS | NW-UCLA |
|---|---|---|---|---|---|---|
| Self-pretrain | 86.6 | 91.0 | 94.0 | 65.8 | 64.5 | 91.6 |
| Universal-pretrain | **87.0** | **91.6** | **95.2** | **71.1** | **68.0** | **93.5** |

Table 9: Universal pretraining evaluation on multiple skeleton datasets.

**Transfer Learning Evaluation Results.** In the transfer learning evaluation protocol, models are first pre-trained on a source dataset and then fine-tuned on a distinct target dataset to assess the generalization of learned representations. Here, PKU-MMD II serves as the target dataset, while NTU-60, NTU-120, and PKU-MMD I are used as source datasets. As shown in Table 7, our method achieves strong performanceacross all transfer settings, demonstrating superior generalization and robustness over existing methods.

## 4.5 BROADER APPLICATIONS

As discussed in Section 3.1, our proposed S2I provides a unified representation for tackling both Cross-Format Transfer Learning and Universal Skeleton Representation Learning. To demonstrate its advantages, we conduct experiments under these two settings.

**Cross-Format Transfer Learning Evaluation Results.** We evaluate S2I on three heterogeneous skeleton datasets: NTU-60 (25 joints), Toyota (13 joints), and NW-UCLA (20 joints). Existing methods often require joint downsampling or interpolation to match skeleton formats, which leads to information loss or noise. In contrast, our S2I approach preserves structural information by converting skeleton sequences into a format-agnostic image representation. As shown in Table 8, S2I achieves clear improvements over prior works. Specifically, 3s-S2I reaches 53.8% on Toyota (CV1), surpassing existing methods by a significant margin.

**Universal Representation Pretraining Evaluation Results.** To further validate the generalizability of our representation, we conduct universal representation learning by aggregating training data from multiple datasets (NTU120-Csub, PKU-I-CS, PKU-II-CS, Toyota-CS, and NW-UCLA). [2] The experiments are performed on joint data. As shown in Table 9, the universal pretraining consistently boosts performance across all evaluated datasets compared to individual dataset self-pretraining.

## 5 CONCLUSION

In this paper, we propose Skeleton-to-Image Encoding (S2I), a simple yet effective representation that bridges skeleton sequences with vision-pretrained models. By reformatting spatial-temporal skeleton data into image-like structures, S2I enables direct use of powerful pretrained vision models without requiring skeleton-specific design modifications. Extensive experiments on five benchmark datasets show that S2I achieves competitive performance in skeleton representation learning, supporting challenging tasks such as cross-format skeleton transfer learning and universal skeleton pretraining. In the future, we plan to extend S2I to other larger vision models (e.g., VLMs and multi-modal vision models) and explore its potential in multi-modal action recognition, including joint modeling with RGB videos and other sensor data.

---

[2]More details of implementations and skeleton partitioning are provided in Appendix A.5.

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

## A APPENDIX

We provide additional materials in the Appendix that could not be included in the main manuscript due to space constraints. The appendix consists of five parts: (1) detailed descriptions of the network architecture used in our experiments (MAE (He et al., 2022) and DiffMAE (Wei et al., 2023)); (2) pretraining strategies applied for skeleton representation learning; (3) downstream evaluation protocols, including fine-tuning and linear probing; (4) a structural comparison between existing pipelines and our proposed method under cross-format and universal settings; (5) implementation details for universal skeleton representation learning, including skeleton partitioning strategies and preprocessing procedures.

### A.1 NETWORK ARCHITECTURE AND CONFIGURATIONS

**MAE (He et al., 2022).** Our MAE implementation follows the asymmetric encoder-decoder architecture proposed in (He et al., 2022). The encoder is a Vision Transformer (ViT-B/16) (Dosovitskiy et al., 2021), where "16" denotes the patch size. It operates only on the visible (unmasked) patches, without using any mask tokens. Each patch is linearly projected and added with fixed sinusoidal positional embeddings. The encoder consists of 12 Transformer blocks, each composed of a multi-head self-attention layer (Vaswani et al., 2017), a feed-forward MLP, and LayerNorm (Ba et al., 2016) applied before both submodules. A standard class token is used during fine-tuning but omitted during pretraining. The decoder is lightweight and receives both the encoded visible patches and learnable mask tokens. It comprises 8 Transformer blocks (Vaswani et al., 2017) with a hidden size of 512 and projects the output back to pixel space to reconstruct the original input. Reconstruction is performed only on masked patches, and the loss is computed as the mean squared error (MSE) between the predicted and ground-truth pixel values.

**DiffMAE (Wei et al., 2023).** Our implementation of DiffMAE follows the asymmetric encoder-decoder architecture proposed in (Wei et al., 2023), using ViT-B/16 as the encoder. The encoder processes only visible patches, and the decoder reconstructs masked regions from noisy inputs sampled through a forward diffusion process. Following the original design, we append a LayerNorm to the encoder output, followed by a linear projection to align feature dimensions with the decoder. Fixed sinusoidal positional embeddings are added to both the encoder and decoder inputs. We do not use relative positional encodings or layer scale. Additionally, separate linear projections are applied to clean and noisy tokens, respectively.

Unlike MAE, which directly regresses pixel values, DiffMAE formulates masked region prediction as a conditional generation task via a diffusion-based denoising process. Specifically, we adopt the *cross decoder* variant, where each noisy token (corresponding to a masked patch) independently attends to encoder outputs via cross-attention, without interacting with other noise tokens. This avoids shortcut paths and promotes more effective encoder pretraining. For the diffusion process, we follow a linear variance schedule (Ho et al., 2020), with the number of timesteps set to $T = 1000$.

### A.2 PRETRAINING SETUP AND STRATEGIES

**MAE (He et al., 2022).** We follow the general pretraining setup of MAE (He et al., 2022), with modifications to accommodate the smaller scale and different nature of skeleton datasets compared to ImageNet. Specifically, we use a batch size of 512 and train for 800 epochs. For data augmentation, we employ a customized pipeline designed for S2I skeleton representation. In contrast to the original MAE, which uses standard image augmentations such as `ColorJitter` and `DropPath`, we adopt lightweight and semantically consistent transformations: random horizontal flip ($p = 0.5$), random rotation, random affine scaling and translation, and additive Gaussian noise with standard deviation. All augmentations are applied before normalization. All other components, including optimizer settings and learning rate schedule, remain consistent with the original MAE. The full configuration is summarized in Table 10.

Table 10: Pretraining settings for MAE on S2I representation.

| Config | Value |
|---|---|
| Optimizer | AdamW |
| Base learning rate | 1.5e-4 |
| Weight decay | 0.05 |
| Optimizer momentum | $\beta_1, \beta_2 = 0.9, 0.95$ |
| Batch size | 512 |
| Learning rate schedule | cosine decay |
| Epochs | 800 |
| Warmup epochs | 40 |
| Augmentation | Horizontal flip (p=0.5), Random rotation, Random affine, Gaussian noise |

**DiffMAE (Wei et al., 2023).** We follow the official DiffMAE pretraining setup on ImageNet, using the same optimizer, learning rate schedule, and number of training epochs (1600). To account for the smaller scale of skeleton dataset, we reduce the batch size to 512. In place of `RandomResizedCrop`, we adopt a skeleton-related augmentation pipeline consisting of horizontal flip, random rotation, affine transformation, and additive Gaussian noise, as detailed above. The full pretraining configuration is summarized in Table 11.

For the diffusion process, we follow the noise schedule formulation in (Wei et al., 2023), where each forward sample $x_t^m$ is reparameterized as:

$$x_t^m = \sqrt{\bar{\alpha}_t} x_0^m + \sqrt{1 - \bar{\alpha}_t} \epsilon,$$

with $\alpha_t = 1 - \beta_t$ and $\bar{\alpha}_t = \prod_{i=1}^t \alpha_i$. To control the noise level, we introduce a hyperparameter $\rho$ following (Wei et al., 2023), which modulates the variance schedule by exponentiating each $\beta_t$ as $\beta_t^\rho$. We set $\rho = 1.0$ by default, which recovers the standard linear schedule in (Ho et al., 2020), where $\beta_t$ increases linearly from $10^{-4}$ to 0.02.

Table 11: Pretraining settings for DiffMAE on S2I representation.

| Config | Value |
|---|---|
| Optimizer | AdamW |
| Weight decay | 0.05 |
| Base learning rate | 1.5e-4 |
| Optimizer momentum | $\beta_1, \beta_2 = 0.9, 0.95$ |
| Batch size | 512 |
| Learning rate schedule | cosine decay |
| Epochs | 1600 |
| Warmup epochs | 40 |
| Augmentation | Horizontal flip (p=0.5), Random rotation, Random affine, Gaussian noise |

## A.3 DOWNSTREAM TRAINING PROTOCOLS: FINE-TUNING AND LINEAR PROBING

We extract features from the encoder output for downstream tasks, including fine-tuning and linear probing. Following the standard ViT design (Dosovitskiy et al., 2021), which includes a class token, we append an

auxiliary dummy token to the input sequence during pretraining, as in (He et al., 2022). This token is treated as the class token and used for classification in both fine-tuning and linear probing.

**Fine-tuning.** Our fine-tuning protocol follows standard supervised ViT training (Dosovitskiy et al., 2021), with simplified configurations tailored for skeleton-based inputs. Specifically, we omit image-based data augmentation and regularization techniques such as RandAugment, label smoothing, mixup, and cutmix. Instead, we apply a skeleton-related augmentation pipeline, including horizontal flip, rotation, affine transformation, and Gaussian noise. We use the AdamW optimizer with a base learning rate of $1e-3$ and a cosine learning rate decay schedule. Layer-wise learning rate decay is applied with a decay rate of 0.75, following (Bao et al., 2021). We set the batch size to 128 for NTU-60, NTU-120, PKU-MMD, and Toyota datasets, and reduce it to 32 for the smaller NW-UCLA dataset to prevent overfitting. The warmup period is fixed to 10 epochs, and training is performed for 100 epochs in total.

Table 12: Fine-tuning settings for S2I representation.

| Config | Value |
|---|---|
| Optimizer | AdamW |
| Base learning rate | 1e-3 |
| Weight decay | 0.05 |
| Layer-wise lr decay | 0.75 |
| Batch size | 128 (NTU/PKU/Toyota), 32 (N-UCLA) |
| Learning rate schedule | cosine decay |
| Warmup epochs | 10 |
| Training epochs | 100 |
| Augmentation | Horizontal flip (p=0.5), Random rotation, Random affine, Gaussian noise |

**Linear probing.** For linear probing, we freeze the encoder and train only a linear classification head on top of the extracted features. We follow a simplified setup without additional regularization such as weight decay, label smoothing, or mixup. The optimizer is SGD with a base learning rate of 0.2, cosine decay schedule, and momentum set to 0.9. A warmup of 10 epochs is applied, and training is conducted for 100 epochs in total. We use the same skeleton-related data augmentation as in fine-tuning. Detailed configurations are listed in Table 13.

Table 13: Linear probing settings for S2I representation.

| Config | Value |
|---|---|
| Optimizer | SGD |
| Base learning rate | 2e-1 |
| Weight decay | 0 |
| optimizer momentum | 0.9 |
| Batch size | 128 |
| Learning rate schedule | cosine decay |
| Warmup epochs | 10 |
| Training epochs | 100 |
| Augmentation | Horizontal flip (p=0.5), Random rotation, Random affine, Gaussian noise |

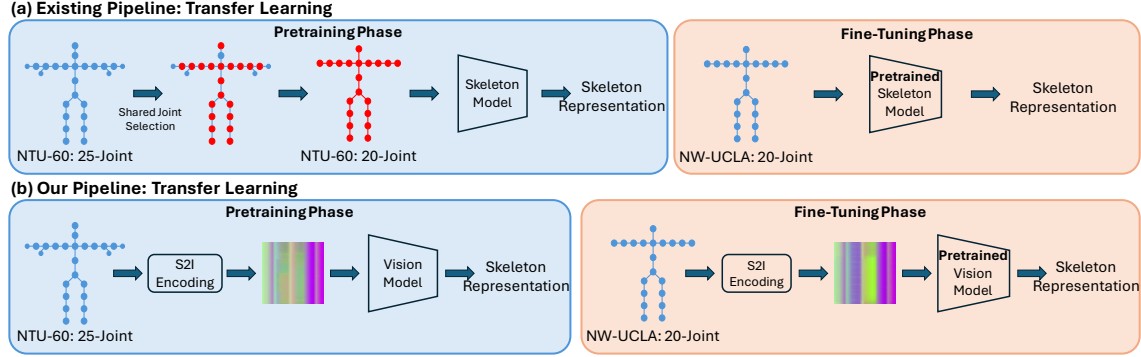

Figure 4: Comparison between the existing skeleton-specific pipeline and our proposed S2I-based pipeline for transfer learning across datasets. (a) Existing methods first align joint formats before pretraining and fine-tuning on target datasets, which may lead to information loss. (b) Our approach bypasses manual joint selection by encoding the raw skeleton sequence into an image via S2I, enabling unified processing across datasets using a vision model.

## A.4 Pipeline Comparison under Cross-Format and Universal Settings

**Cross-Format Representation Learning.** To evaluate cross-format transfer performance, we compare our pipeline against the conventional transfer learning paradigm, using the transfer from NTU-60 to NW-UCLA as an example. As illustrated in Figure 4, existing approaches typically begin by manually downsampling the NTU-60 skeleton data (25 joints) to a shared 20-joint subset. A model is then pretrained on this reduced-format data and fine-tuned on NW-UCLA, which naturally adopts the 20-joint format. This process requires explicit joint alignment and may introduce structural mismatches, potentially degrading the quality of learned representations.

In contrast, our S2I-based pipeline removes the need for manual joint selection. We directly encode the raw skeleton sequences—regardless of their joint format—into semantically structured image-like representations. Both NTU-60 and NW-UCLA inputs are processed uniformly by the same visual backbone without any format-specific adaptation. This design enables seamless cross-format transfer while preserving the structural integrity of the original data. Figure 4 provides a visual comparison of the two pipelines, highlighting the simplicity and universality of our approach for cross-format transfer learning.

**Universal Representation Pretraining.** To facilitate universal skeleton representation learning across diverse datasets, we compare our unified pipeline with traditional skeleton-specific approaches. As shown in Figure 5, conventional methods require designing and training separate models tailored to each joint format (e.g., 25-joint, 20-joint, 13-joint), resulting in limited scalability.

In contrast, our approach leverages the S2I (Skeleton-to-Image) encoding to transform skeleton sequences with arbitrary joint formats into structured image representations. These representations are then processed by a shared backbone model, enabling consistent and format-agnostic feature learning. This unified design eliminates the need for joint-level alignment or model reconfiguration, allowing all skeleton datasets—regardless of their original format—to contribute to a single pretraining framework. The resulting representation is thus inherently universal, capable of generalizing across different skeleton domains with no structural compromises.

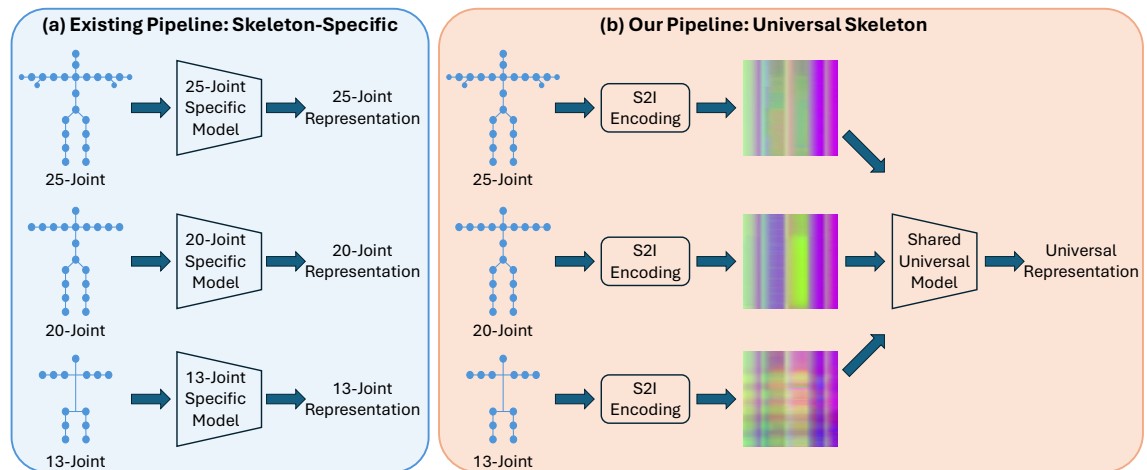

Figure 5: Comparison of skeleton-specific and universal representation learning pipelines. (a) Conventional methods require format-specific models for each skeleton format. (b) Our method encodes arbitrary skeleton formats into image representations via S2I, enabling unified pretraining with a single backbone model.

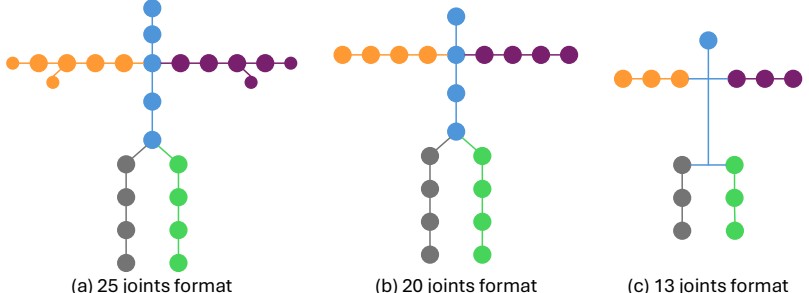

Figure 6: Visualization of three commonly used skeleton formats: (a) 25-joint (NTU/PKU), (b) 20-joint (NW-UCLA), and (c) 13-joint (Toyota). Each skeleton is partitioned into five semantic body parts—Spine, Left Arm, Right Arm, Left Leg, and Right Leg—highlighted in different colors to ensure consistent representation across formats.

## A.5 UNIVERSAL REPRESENTATION PRETRAINING SETUPS

**Implementation Details.** To conduct the Universal Representation Pretraining experiments, we utilize the training splits from multiple datasets, including NTU120 (C-sub) (Liu et al., 2020), PKU-MMD I (CS) (Liu et al., 2020), PKU-MMD II (CS), Toyota-Smarthome (CS) (Das et al., 2019), and NW-UCLA (Wang et al., 2014). Since the NTU60-C-sub (Shahroudy et al., 2016) training set is a subset of NTU120-C-sub, it is not explicitly included.

To ensure consistent input distribution across datasets, we compute the mean and standard deviation of the S2I RGB representations from the NTU120 C-sub training split, and use them to normalize the S2I inputs from all other datasets for scale alignment and stable optimization. Additionally, due to the increased size of the combined training data, we increase the batch size from 512 to 768, while keeping all other training

Table 14: Semantic body part partitioning across different skeleton formats, using joint names.

| Body Part | NTU/PKU (25 joints) | NW-UCLA (20 joints) | Toyota (13 joints) |
|---|---|---|---|
| Spine | head, neck, spine, middle of spine, base of spine | head, spine, middle of spine, base of spine | head |
| Left Arm | left shoulder, left elbow, left wrist, left hand, left thumb, tip of left hand | left shoulder, left elbow, left wrist, left hand | left shoulder, left elbow, left wrist |
| Right Arm | right shoulder, right elbow, right wrist, right hand, right thumb, tip of right hand | right shoulder, right elbow, right wrist, right hand | right shoulder, right elbow, right wrist |
| Left Leg | left hip, left knee, left ankle, left foot | left hip, left knee, left ankle, left foot | left hip, left knee, left ankle |
| Right Leg | right hip, right knee, right ankle, right foot | right hip, right knee, right ankle, right foot | right hip, right knee, right ankle |

configurations unchanged. All datasets are jointly trained in a unified manner, rather than using any sequential or curriculum-based strategy.

**Skeleton Partitioning for Different Formats.** To support universal skeleton representation learning across datasets with different joint formats, we consider three commonly used skeleton configurations, as shown in Figure 6. The NTU-60, NTU-120, and PKU-MMD datasets adopt the 25-joint layout extracted by Kinect V2. NW-UCLA provides 20-joint skeletons from Kinect V1, while the Toyota dataset offers 13-joint skeletons estimated using LCRNet (Rogez et al., 2019).

To enable format-invariant learning, we partition the skeleton into five consistent body parts: *Spine*, *Left Arm*, *Right Arm*, *Left Leg*, and *Right Leg*. Figure 6 color-codes these body parts across the three skeleton formats, and Table 14 lists the corresponding joint names for each part.

