# OpenReview forum: "Skeleton-to-Image Encoding: Enabling Skeleton Representation Learning via Vision-Pretrained Models"
_ICLR.cc/2026/Conference — ICLR 2026 Conference Withdrawn Submission_

### Official Review · Reviewer_fLTB · 2025-10-29

**Soundness:** 2
**Presentation:** 2
**Contribution:** 2
**Rating:** 2
**Confidence:** 3

**Summary:**

This paper focuses on representation learning for 3D skeleton sequences. It achieves this by first converting the skeleton sequences to images, using rgb colours to represent the 3D location of skeleton joint. Specific portions of the image width encode different joints, while the height to encode the temporal evolution over the sequence. Using an image allows reuse of powerful image representation models, e.g. MAE, DiffMAE, which are finetuned on their original pretraining task with the skeleton images. The resulting representation is then finetuned for the skeleton action recognition downstream task.

**Strengths:**

- The idea of leveraging large-scale visual models for skeleton-based representation learning is interesting.
***

- The paper explicitly considers whether the MAE masking strategy should be adapted for skeleton data, although finds that random masking still performs best.
***

- Table 1 shows clear gains from image pretraining, with around +5% for linear probing and +2% for finetuning, indicating that the approach provides tangible representation benefits.
***

- The method achieves strong semi-supervised performance (Table 6), suggesting good transferability under limited-label settings.

**Weaknesses:**

- Related work coverage
- While no current sota skeleton action representation learning methods have considered converting skeleton sequences to images, this seems to have been a common approach in learning from skeleton sequences previously e.g. [D, E, F, G, H, I]
- The presence of this prior work is not necessarily a reason to reject as there is value in bringing ideas back and making them work in new forms, however they should be acknowledged in the related work and it should be determined whether the proposed skeleton to image conversion is equivalent to any of these and whether it offers advantages over any of these previously proposed skeleton-to-image conversions
***

- Experimental Results
- The paper would benefit from stronger positioning relative to recent masked prediction models, especially SkeletonMAE [A], MaskCLR [B], and S-JEPA [C], which are notably absent from comparisons
- The proposed method is not state-of-the-art in linear evaluation, and its relative advantage primarily appears in the semi-supervised setting.
- Without the 3-stream version, the method performs below prior works such as MAMP and MacDiff, and it is unclear whether those methods would also benefit similarly from a 3-stream configuration.
- Backbone clarity is lacking. Methods like MAMP and MacDiff use transformers, while others primarily use graph networks, making direct comparisons less meaningful without explicit specification and increases the importance of comparison to [A, B, C]
***

- Representation Design and Analysis
- The paper would benefit from ablation studies examining the ordering and grouping choices in the skeleton-to-image transformation. It remains unclear whether the particular encoding scheme is optimal or arbitrary.
- This is particularly important in light of the previous works [D-I] and clarifying the advantage over these skeleton-to-image conversions
***

[A] Yan et al. SkeletonMAE: Graph-based Masked Autoencoder for Skeleton Sequence Pre-training. ICCV 2023.
***
[B] Abdelfattah et al. MaskCLR: Attention-Guided Contrastive Learning for Robust Action Representation Learning. CVPR 2024.
***
[C] Abdelfattah et al. S-JEPA: A Joint Embedding Predictive Architecture for Skeletal Action Recognition. ECCV 2024.
***
[D] Mokhtari et al. Human Activity Recognition: A Spatio-temporal Image Encoding of 3D Skeleton Data for Online Action Detection. VISAPP 2022.
***
[E] Enhanced Spatio- Temporal Image Encoding for Online Human Activity Recognition. ICMLA 2023.
***
[F] Li et al. Skeleton Based Action Recognition Using Translation-Scale Invariant Image Mapping And Multi-Scale Deep CNN. ICME Workshops 2017.
***
[G] Li et al. Skeleton-based Action Recognition with Convolutional Neural Networks. ICME Workshops 2017.
***
[H] Caetano et al. Skeleton Image Representation for 3D Action Recognition based on Tree Structure and Reference Joints. SIBGRAPI 2019.
***
[I] Chen et al. Action Recognition with Domain Invariant Features of Skeleton Image. AVSS 2021.

**Questions:**

- Does the proposed skeleton-to-image encoding fundamentally differ from prior skeleton image mapping approaches [D-I]?

- Why are SkeletonMAE, MaskCLR, and S-JEPA omitted from comparison, given their strong relevance as masked prediction models for skeleton data?

- Would MAMP or MacDiff similarly benefit from a 3-stream version?

- Have you explored ablation studies on joint ordering or grouping in the image representation to verify that the current configuration is optimal?

- How sensitive is the method to the specific joint ordering and grouping used in the image representation?

---

### Official Review · Reviewer_GH6A · 2025-11-01

**Soundness:** 3
**Presentation:** 3
**Contribution:** 4
**Rating:** 6
**Confidence:** 4

**Summary:**

1. This paper introduces a skeleton-based image representation to leverage vision-pretrained models.

2. To achieve this, the authors propose a skeleton-to-image encoding approach.

3. The framework can be applied to heterogeneous skeleton representations.

**Strengths:**

1. The proposed method can be applied to heterogeneous skeleton representations, enabling the expansion of the training dataset.

2. Cross-dataset finetuning or universal representation pretraining could also be explored, similar to approaches in image-based action recognition.

3. The skeleton-to-image encoding effectively captures both temporal and spatial information, which is essential for accurate action recognition.

4. The method demonstrates strong performance in low-label regimes, as evidenced by the semi-supervised learning results in Table 6.

**Weaknesses:**

1. Without incorporating additional modalities such as bone or motion features, the performance does not surpass existing methods.

2. The explanation of the bone and motion modalities is insufficient. How are these modalities derived in the proposed framework?

**Questions:**

1. In Section 3.3, the terms block masking and group masking do not match the naming in Table 2. Please ensure consistent terminology.

2. In Table 1, for the Linear-Probe setting without pretrained image and skeleton encoders, are the encoders randomly initialized and only the recognition head trained?

---

### Official Review · Reviewer_vb8h · 2025-11-01

**Soundness:** 2
**Presentation:** 3
**Contribution:** 2
**Rating:** 4
**Confidence:** 3

**Summary:**

This work converts sparse 3D skeleton sequences into a dense image-like format, making them compatible with vision models and enabling effective transfer of visual knowledge to the skeleton domain. It further provides a representation that is agnostic to specific skeleton formats, achieving competitive results across multiple benchmarks and experimental settings.

**Strengths:**

1. Converting the set of keypoints into an image representation is an interesting attempt;

2. The overall writing is clear and easy to follow;

3. The method outperforms state-of-the-art approaches on multiple benchmarks (e.g., NTU-60, NTU-120, PKU-MMD) and shows strong generalization in cross-format and few-shot scenarios;

**Weaknesses:**

1. Mapping the temporal dimension directly to the image height is a simple and straightforward stacking approach. However, this design disrupts temporal continuity. In an image, adjacent rows (i.e., consecutive time frames) are spatially continuous, whereas in skeleton sequences, the same joint across consecutive frames appears in different rows, and different joints within the same row may be far apart in physical space (e.g., left hand vs. right foot). Such a representation may violate the locality assumption that visual models (e.g., ViT) rely on. A potentially more natural representation would be to treat each frame’s skeletal pose as a small image patch, then arrange these patches sequentially over time.

2. The RGB channels of an image typically represent color information, whose numerical distribution and semantics differ greatly from 3D spatial coordinates. Directly using coordinates as pixel values may lead to a distributional shift that pretrained vision models on natural images (e.g., ImageNet) are not adapted to handle. Moreover, it is unclear whether the authors applied any normalization. If raw coordinates were used, their value range might be inconsistent with the expected pixel range (usually 0–255), potentially affecting model stability and performance.

3. For inherently sparse skeleton data, forcibly upsampling them into high-resolution images may result in most pixels being generated by interpolation, carrying very limited information, while the true signal (original joint coordinates) occupies only a small portion. This essentially introduces a large amount of low-information or even noisy interpolated data around the valid signal. Therefore, it would be necessary to compare different interpolation strategies to justify the chosen one.

4. The use of large-scale visual models such as ViT-B/16 may introduce substantial computational and memory overhead. The current version of the paper does not seem to include a comparison of computational efficiency, which would be important to evaluate the practicality of the proposed approach.

**Questions:**

Please See the Weaknesses.

---

### Official Review · Reviewer_i9Nz · 2025-11-01

**Soundness:** 2
**Presentation:** 2
**Contribution:** 2
**Rating:** 2
**Confidence:** 5

**Summary:**

This paper proposes a new method for self-supervised skeleton action recognition. Firstly, it converts the skeleton sequences into image-like data. Then, it fine-tunes the pre-trained vision models using MAE techniques on these skeleton-based images for self-supervised training. Experimental results demonstrate competitive performance across several benchmark datasets.

**Strengths:**

1. The experiments show that fine-tuning pre-trained vision models on image-like skeleton data can achieve competitive results, suggesting that visual semantics may share correlations with skeleton motion semantics.
2. Converting skeleton sequences into image-like representations effectively addresses the structural discrepancy among different skeleton types.

**Weaknesses:**

1. The motivation for using pre-trained vision models in self-supervised skeleton action recognition is unclear and insufficiently explained. Moreover, the necessity of applying this paradigm to skeletal data is not well justified. From the performance improvement perspective, the transferred knowledge from vision models appears limited. For example, the performance under linear evaluation is worse than in prior works. This indicates that the learned skeleton representations from pre-trained vision models are inferior to those from skeleton-specific designs trained from scratch.
2. Distilling or transferring knowledge from pre-trained vision-language models (VLMs) for self-supervised skeleton action recognition has been explored in several prior works [1,2], which should be properly discussed in the related work section. Furthermore, existing studies have shown that distilling video or text knowledge from VLMs into skeleton representations can be effective, which further limits the novelty of this work.
3. The transformation of skeleton data into image-like formats has been explored many years ago [3,4,5,6], particularly with CNN-based feature extractors. These works are not well introduced or discussed in the related work section. In contrast, the present study only fine-tunes pre-trained vision models on image-like skeleton data instead of training CNNs from scratch, which provides a rather limited contribution since such transformations have been thoroughly investigated. In addition, more recent works addressing cross-format skeleton representation should also be introduced and fairly compared [7,8,9].
4. The experimental section lacks qualitative analysis of the proposed method, such as t-SNE visualizations of the learned feature distributions, which is necessary to evaluate the representation quality for self-supervised skeleton representation learning.

[1] Sinha A, Reilly D, Bremond F, et al. SKI Models: Skeleton Induced Vision-Language Embeddings for Understanding Activities of Daily Living[C]//Proceedings of the AAAI Conference on Artificial Intelligence. 2025, 39(7): 6931-6939.

[2] Chen Y, He T, Fu J, et al. Vision-language meets the skeleton: Progressively distillation with cross-modal knowledge for 3d action representation learning[J]. IEEE Transactions on Multimedia, 2024.

[3] Li C, Wang P, Wang S, et al. Skeleton-based action recognition using LSTM and CNN[C]//2017 IEEE International conference on multimedia & expo workshops (ICMEW). IEEE, 2017: 585-590.

[4] Ding Z, Wang P, Ogunbona P O, et al. Investigation of different skeleton features for cnn-based 3d action recognition[C]//2017 IEEE International conference on multimedia & expo workshops (ICMEW). IEEE, 2017: 617-622

[5] Li B, Dai Y, Cheng X, et al. Skeleton based action recognition using translation-scale invariant image mapping and multi-scale deep CNN[C]//2017 IEEE International Conference on Multimedia & Expo Workshops (ICMEW). IEEE, 2017: 601-604.

[6] Li C, Zhong Q, Xie D, et al. Skeleton-based action recognition with convolutional neural networks[C]//2017 IEEE international conference on multimedia & expo workshops (ICMEW). IEEE, 2017: 597-600.

[7] Mo C, Hu K, Long C, et al. Motion Keyframe Interpolation for Any Human Skeleton via Temporally Consistent Point Cloud Sampling and Reconstruction[C]//European Conference on Computer Vision. Cham: Springer Nature Switzerland, 2024: 159-175.

[8] Mo C A, Hu K, Long C, et al. PUMPS: Skeleton-agnostic point-based universal motion pre-training for synthesis in human motion tasks[C]//Proceedings of the IEEE/CVF International Conference on Computer Vision. 2025: 14496-14506.

[9] Liu H, Li Y, Mu T J, et al. Recovering complete actions for cross-dataset skeleton action recognition[J]. Advances in Neural Information Processing Systems, 2024, 37: 92055-92081.

**Questions:**

1. In Section 3.1, how is a human–human interaction skeleton sequence converted into an image-like data? Since several datasets include two-person interaction actions, this should be clearly described.
2. In the ablation study, there is no analysis of how the joint order design (lines 157–159) influences the final performance.
3. Regarding the model architecture, both MAE and DiffMAE are based on ViT-B. However, prior self-supervised skeleton methods commonly use GRU or ST-GCN backbones to demonstrate their effectiveness. Does the stronger representation capacity of ViT-B lead to higher performance here? If so, the comparison may be somewhat unfair.
4. In Table 1, what does "skeleton pretrain" refer to? The term "image pretrain" clearly denotes fine-tuning MAE (or DiffMAE) pre-trained models, but there is no clear explanation of what "skeleton pretrain" means in the paper.

---

### Official Review · Reviewer_ufUd · 2025-11-01

**Soundness:** 2
**Presentation:** 3
**Contribution:** 2
**Rating:** 4
**Confidence:** 2

**Summary:**

The topic of 3D human skeleton-based analysis is highly promising, especially given the rapid progress of large language models, which further expands its potential application space. This paper addresses key challenges such as the fundamental discrepancy in data formats and the scarcity of available datasets. To tackle these issues, the authors propose S2I, a framework designed to convert skeleton sequences into image-like representations. Notably, S2I demonstrates strong practicality, as it effectively transforms skeleton data into a unified image-based format while accommodating heterogeneous data sources. In addition, the authors conduct extensive experiments across different datasets and evaluation settings, which provide convincing evidence of the effectiveness of the proposed approach.

**Strengths:**

S1: The proposed skeleton-to-image encoding exhibits strong generality. It reformats sparse 3D skeletal data into an image-like representation, providing a novel and useful form of data representation.

S2: The paper is clearly written, and the descriptions of the experimental settings and implementation details are well-articulated.

S3: The experimental section is relatively thorough and validates the effectiveness of the proposed method. It is worth noting that the method does not require any task-specific architectural modifications to the skeletal data, which is beneficial in practical applications.

**Weaknesses:**

W1: While the proposed skeleton-to-image encoding provides a useful and generalizable way to decouple skeletal representation from dataset-specific joint configurations, the contribution may be regarded as more of a technical refinement than a conceptual innovation. Nonetheless, it does offer practical value and may inspire further exploration in this direction.

W2: Both MAE and DiffMAE used in this paper are based on the ViT-B architecture, which is relatively modest in scale. Considering the dominant role of large-scale language models and foundation models in recent years, it would be valuable to evaluate whether larger or more advanced models could further enhance the performance of the proposed method. Testing the effectiveness of the approach under such models would provide a more comprehensive validation.

W3: The baselines used in the experiments are relatively outdated; it would be beneficial to consider more recent methods for comparison.

**Questions:**

Q1: Can the proposed encoding method be transferred to larger-scale models?

---

### Official Review · Reviewer_TeQL · 2025-11-01

**Soundness:** 3
**Presentation:** 3
**Contribution:** 3
**Rating:** 4
**Confidence:** 4

**Summary:**

This paper introduces Skeleton-to-Image Encoding (S2I), a method that transforms 3D skeleton sequences into image-like representations by mapping joint coordinates to RGB channels, partitioning joints into semantic body parts, and resizing to standard image dimensions. This encoding enables the direct application of pre-trained vision models (MAE, DiffMAE) to skeleton-based action recognition without specialized architectures. The method demonstrates strong performance across multiple benchmarks (NTU-60, NTU-120, PKU-MMD, NW-UCLA, Toyota) and shows particular strength in cross-format generalization and universal representation learning by handling heterogeneous skeleton structures through a unified image-based representation.

**Strengths:**

+ The core idea of converting sparse skeleton data into a dense image format is novel. The S2I representation demonstrates good adaptability to different skeleton definitions.

+ The experiments are thorough, covering five diverse datasets, including the challenging real-world Toyota dataset. The method is rigorously evaluated under multiple settings (self-supervised learning, linear evaluation, fine-tuning, semi-supervised learning, transfer learning, cross-format transfer), demonstrating robust and superior performance.

+ The approach is conceptually straightforward and directly compatible with mainstream vision models (MAE, DiffMAE), leveraging their powerful pre-trained weights without requiring task-specific architectural modifications.

**Weaknesses:**

- The S2I transformation is heuristic. The paper provides little theoretical analysis on how this specific encoding preserves the spatial structure information of the skeleton or its impact on action semantics. The principles behind the body-part ordering and the interpolation method are not sufficiently justified.

- The direct mapping of 3D coordinates to the RGB domain is a key design choice. The authors should investigate and discuss whether this is the optimal mapping or if alternative encodings (e.g., using different color spaces or channel assignments) could yield better performance or representational fidelity.

- A critical weakness is the lack of interpretability. After mapping skeleton data to the image domain, it remains unclear which regions of the generated "image" are most critical for action recognition. The authors should provide visualization analyses (e.g., attention maps from the ViT, Grad-CAM-like techniques) to identify the salient regions and explain what the vision model is actually "seeing" and leveraging in these synthetic images.

- The paper does not provide a failure analysis or a clear discussion of limitations. On some metrics, the method does not achieve state-of-the-art (SOTA). The authors should analyze these cases: What types of actions is the model particularly good or bad at? For instance, does the method struggle with highly dynamic motions where fine-grained temporal information might be lost during resizing? A discussion of the method's boundaries and typical failure cases is essential.

- The reliance on existing, off-the-shelf vision models places the work in the category of "effective engineering." The core contribution is the input representation, and its optimality and generality across different vision backbones (e.g., ConvNeXt, CLIP) remains unverified.

- The computational cost of using ViT-B models pre-trained on ImageNet is not discussed. A comparison of training time, inference speed, and parameter count against efficient skeleton-specific models (e.g., GCNs) is crucial for assessing the practical utility.

Overall, the core idea is interesting and the empirical results are extensive. However, theoretical depth and interpretability analysis are needed. If the authors could address the concerns, I would be willing to upgrade my score.

**Questions:**

Could you provide visualizations (e.g., attention maps) to show which parts of the S2I "image" are most influential for the model's decisions? This would greatly help in understanding what the vision model learns from this representation.

---

### Official Review · Reviewer_LzUg · 2025-11-11

**Soundness:** 3
**Presentation:** 3
**Contribution:** 2
**Rating:** 6
**Confidence:** 3

**Summary:**

The paper proposes Skeleton-to-Image Encoding (S2I), which converts 3D skeleton sequences into image-like representations to enable vision-pretrained models (e.g., MAE, DiffMAE) for skeleton representation learning. This unified encoding bridges the gap between skeleton and visual domains, supporting cross-format and universal skeleton pretraining without skeleton-specific architectures. Extensive experiments on multiple benchmarks demonstrate consistent but incremental improvements over existing self-supervised and transfer learning baselines.

**Strengths:**

1. Conceptual simplicity and reusability: The S2I design elegantly reuses pretrained vision transformers for skeleton data without requiring task-specific architectural redesigns.

2. Strong generalization: Demonstrates robustness in cross-format and transfer-learning scenarios, supporting heterogeneous skeleton datasets with a unified encoding.

3. Comprehensive empirical validation: Includes detailed ablations on masking strategies, modalities, and pretrained initialization across five datasets.

**Weaknesses:**

1. Overlap with prior art: **Many earlier works (e.g., Skepxels, Translation-Scale Invariant Image Mapping) have already transformed skeletons into image-like forms for CNN-based feature extraction using pretrained image models**. The paper should clearly articulate what S2I contributes beyond these established skeleton-to-image CNN paradigms.   **Repeated or incorrect phrases like “for the first time”**, in this regard.

2. Incremental performance gains: The improvements over strong recent baselines appear modest, suggesting that the contribution may be more engineering-oriented than conceptual.

3. Lack of intermediate justification: The paper would benefit from analyses showing why the generated image-like representations improve learning—e.g., intermediate visualizations, feature-space comparisons, or ablation on encoding components.

4. Assumption of full visual transferability: The 2D projection may lose temporal and kinematic details intrinsic to motion, raising doubts about its biological interpretability.

5. Computational overhead: Training large ViT-based models (MAE, DiffMAE) for small skeleton datasets is resource-intensive, with uncertain scalability to real-world motion data.

**Questions:**

N/A

---

### Note · Authors · 2025-11-18

I have read and agree with the venue's withdrawal policy on behalf of myself and my co-authors.